# Seasonal plasticity in GABA$_A$ signaling is necessary for restoring phase synchrony in the master circadian clock network

**Kayla E Rohr[†], Harshida Pancholi[†], Shabi Haider, Christopher Karow, David Modert, Nicholas J Raddatz, Jennifer Evans***

Department of Biomedical Sciences, Marquette University, Milwaukee, United States

**Abstract** Annual changes in the environment threaten survival, and numerous biological processes in mammals adjust to this challenge via seasonal encoding by the suprachiasmatic nucleus (SCN). To tune behavior according to day length, SCN neurons display unified rhythms with synchronous phasing when days are short, but will divide into two sub-clusters when days are long. The transition between SCN states is critical for maintaining behavioral responses to seasonal change, but the mechanisms regulating this form of neuroplasticity remain unclear. Here we identify that a switch in chloride transport and GABA$_A$ signaling is critical for maintaining state plasticity in the SCN network. Further, we reveal that blocking excitatory GABA$_A$ signaling locks the SCN into its long day state. Collectively, these data demonstrate that plasticity in GABA$_A$ signaling dictates how clock neurons interact to maintain environmental encoding. Further, this work highlights factors that may influence susceptibility to seasonal disorders in humans.
DOI: https://doi.org/10.7554/eLife.49578.001

**\*For correspondence:**
jennifer.evans@marquette.edu

[†]These authors contributed equally to this work

**Competing interests:** The authors declare that no competing interests exist.

## Introduction

Daily rhythms generated by the circadian system serve to anticipate changes in the environment caused by the Earth's rotation (*Hut and Beersma, 2011*; *Pittendrigh, 1960*). In mammals, the circadian system is a hierarchical collection of biological clocks orchestrated by a master pacemaker in the suprachiasmatic nucleus (SCN) of the anterior hypothalamus (*Mohawk et al., 2012*). The SCN itself is a network of clock cells that interact with one another to regulate emergent circuit properties, process photic cues, and provide daily signals to downstream tissues (*Evans, 2016*; *Hastings et al., 2018*). At the cellular level, SCN neurons display ca. 24 hr rhythms in electrical and molecular activity (*Hastings et al., 2018*) generated by genetic feedback loops that control daily expression of clock proteins (*Buhr and Takahashi, 2013*) and the clock-controlled genes that regulate cellular physiology (*Zhang et al., 2014*). Although the circadian molecular clock drives cellular rhythms, intercellular interactions among SCN neurons synchronize, amplify, and stabilize rhythms of cells within the network (*Evans, 2016*; *Hastings et al., 2018*). In addition, the SCN adjusts to the environment, effectively matching the 24 hr period of the solar cycle, the phase of the local time zone, and the waveform of the prevailing day length as it changes over the year. Thus, the SCN network serves as both daily clock and annual calendar, and defining the circuit-level mechanisms by which SCN neurons interact is paramount for understanding the temporal regulation of behavior and physiology.

Seasonal changes in the environment threaten the survival and well being of organisms living on this planet. To cope with this challenge, many biological processes are modulated on a seasonal basis. Although daily timekeeping is a cellular property, seasonal encoding is accomplished via changes in the spatiotemporal relationships of SCN clock cells (*Meijer et al., 2010*; *Evans and*

**eLife digest** In winter, as the days become shorter, millions of people find that their mood and energy levels start to drop. They crave carbohydrates, struggle with their weight, and find it harder to get out of bed in the mornings. These individuals are suffering from the 'winter blues' or seasonal affective disorder (SAD), and most find that their symptoms spontaneously improve in the spring when the days become longer again. Many also benefit from bright light therapy during the winter months, but not everyone responds fully to this treatment, so additional options are needed.

The winter blues occur when the brain adjusts to changes in day length with the onset of winter. The brain region responsible for making this adjustment is the suprachiasmatic nucleus (SCN). The SCN is the master clock of the brain that coordinates the body's circadian rhythms – the daily fluctuations in things like appetite, body temperature, sleep and wakefulness.

But as well as being the brain's clock, the SCN is also the brain's calendar. In winter, when the days are short, SCN neurons coordinate their activity and fire in synchrony. But in summer, when the days are long, SCN neurons divide into two clusters, which fire at different times. By transitioning between these two states, the SCN helps the body adjust to seasonal changes in day length. Rohr, Pancholi et al. now provide new insight into the mechanism behind this process by showing that light alters the neurochemistry of the SCN.

Exposing mice to long days causes a brain chemical called GABA to switch from inhibiting neurons in the SCN to activating them. Blocking this switch from inhibition to activation locks the SCN into its 'summer state'. Rohr, Pancholi et al. propose that this failure to transition to the winter state may be an interesting way to prevent the winter blues.

While much remains to be learned about this process, these findings pave the way for better understanding the neurobiology of winter depression and how best to treat it.

DOI: https://doi.org/10.7554/eLife.49578.002

Gorman, 2016). When days are short, SCN neurons form a highly synchronized population of cellular clocks with similar times of daily protein expression and electrical activity. In contrast, when days are long, the SCN network switches to an alternate state characterized by two subclusters of clock neurons that cycle in anti-phase (Evans et al., 2013; Inagaki et al., 2007). The ability of the SCN network to encode season is critical for the regulation of a large variety of biological processes, including reproduction, immune function, and metabolism. In humans, seasonality likewise influences a wide range of functions such as sleep, metabolism, and neurotransmission (Garbazza and Benedetti, 2018), which can produce annually recurrent pathology in 3–10% of people (Wehr et al., 2001; Lewy et al., 2006). Although seasonal affective disorder is closely linked to photic modulation of circadian clock function (Wirz-Justice, 2018), the mechanisms that regulate sensitivity to day length remain ill-defined.

GABA is the primary neurotransmitter expressed by SCN neurons (Albers et al., 2017), and recent work indicates that day length modulates GABA signaling in the SCN network (Evans et al., 2013; Myung et al., 2015; Farajnia et al., 2014). Although GABA is typically classified as an inhibitory neurotransmitter, the GABA$_A$ receptor is a heteropentameric ligand-gated ion channel permeable to chloride and bicarbonate that can elicit either hyperpolarization or depolarization depending on the electrochemical gradient of chloride (Kaila et al., 2014). In mature neurons, levels of intracellular chloride are regulated by the Cl$^-$ extruder KCC2 (K$^+$ Cl$^-$ co-transporter 2) and the Cl$^-$ importer NKCC1 (Na$^+$ K$^+$ Cl$^-$ co-transporter 1). Changes in the relative expression and/or function of these two chloride co-transporters cause ionic plasticity: short- and long-term modulation that alters neuronal responses to GABA$_A$ signaling (Kaila et al., 2014). The most well documented example of this neuroplasticity occurs during development (Rivera et al., 1999), when NKCC1 drives a depolarizing chloride reversal potential in immature neurons, but up-regulation of KCC2 causes a switch to the hyperpolarizing GABA response in adulthood. However, this process is not immutable; with downregulation of KCC2 in mature neurons causing a switch to excitatory GABA$_A$ signaling in several distinct circuits in the adult brain (Chung, 2012; Hewitt et al., 2009; Lee et al., 2011; Ostroumov et al., 2016; Sarkar et al., 2011). In the SCN, recent work has shown that exposure to long days increases the number of neurons that respond to GABA with depolarization

(*Farajnia et al., 2014*) and elevates intracellular chloride (*Myung et al., 2015*). Further, alterations in GABA$_A$ signaling are associated with changes in circadian behavior (*Myung et al., 2015*; *Farajnia et al., 2014*; *DeWoskin et al., 2015*), and the role of GABA$_A$ signaling in coupling SCN neurons depends on seasonal state (*Evans et al., 2013*). Collectively, this work suggests that GABA$_A$ signaling in the SCN varies with day length, but it remains unclear how plasticity in GABA circuits regulates seasonal encoding by the master clock network.

Here we provide insight into the mechanisms and functional significance of seasonal plasticity in GABA$_A$ signaling in the SCN network. First, we show how changes in day length modulate molecular responses of the SCN clock to GABA stimulation. Next, we provide insight into mechanisms underlying seasonal plasticity in GABA$_A$ signaling by demonstrating that long days decrease KCC2 expression in the specific SCN compartment that processes light. Lastly, we test the functional significance of photoperiodic changes in chloride transport and GABA$_A$ signaling for SCN network dynamics. Our results demonstrate that the seasonal switch in GABA$_A$ signaling accelerates recovery of the SCN network back to its synchronous state. Through pharmacological interrogation, we define the regional locus of this neuroadaptation, identify consequences for network encoding, and shed new light on how this form of plasticity may modulate daily rhythms in behavior. Overall, these results reveal a novel mechanism by which seasonal adjustments in GABA$_A$ circuits of the master clock serve to maintain sensitivity to changes in day length.

## Results

### Long days modulate molecular responses of the SCN clock to GABA stimulation

To gain greater insight into the functional consequences of seasonal changes in GABA$_A$ signaling, we first tested how day length modulates molecular responses of the SCN clock to GABA. Previous work has predicted that the seasonal switch to excitatory GABA$_A$ signaling will alter how the SCN clock is reset by tonic GABA stimulation (*DeWoskin et al., 2015*). To test this prediction, we examined photoperiodic changes in GABA-induced resetting using PER2::LUC rhythms as a readout of the SCN molecular clock. mPer2$^{Luc}$ mice were exposed to either 12 hr of light per day (L12) or a long-day with 20 hr of light per day (L20) for at least 8 weeks. SCN slices were collected from mice housed under each photoperiod, and PER2::LUC rhythms were monitored for three cycles in vitro before treatment with either 200 µM GABA or vehicle (*Figure 1—figure supplement 1*). GABA was not washed out following application to provide a tonic stimulus, and its concentration was stable in culture for at least 24 hr (*Figure 1—figure supplement 1*). As in previous work (*Evans et al., 2013*), the SCN clock shifted to a later phase when GABA was applied to L12 slices at the trough of the PER2::LUC rhythm (*Figure 1—figure supplement 1*). Importantly, this resetting response was blocked by the GABA$_A$ receptor antagonist bicuculline (*Figure 1—figure supplement 1*), consistent with a prior study demonstrating that GABA-induced resetting is mediated by GABA$_A$ signaling and not GABA$_B$ signaling (*Liu and Reppert, 2000*).

Next we examined whether long days altered SCN responses to GABA by measuring resetting responses at times spanning the circadian cycle (*Figure 1A*). In L12 slices, GABA shifted the PER2::LUC rhythm in a phase-dependent manner, eliciting phase delays during early subjective night and phase advances during other times of the circadian cycle (*Figure 1A–C*). Notably, the shape of this phase-resetting rhythm was similar to that reported previously for GABA-induced phase shifts of electrical rhythms in dissociated SCN neurons (*Liu and Reppert, 2000*). In L20 slices, GABA elicited phase delays for a larger proportion of the circadian cycle and produced phase advances at later times relative to L12 slices (*Figure 1A–C*). These results indicate that the waveform of the GABA-induced resetting rhythm is altered by photoperiod, as illustrated by either a phase response curve (*Figure 1A*) or polar plot (*Figure 1B*). On the other hand, long day exposure did not alter systematically the magnitude of resetting to this dose of GABA (*Figure 1C*) or vehicle (*Figure 1—figure supplement 1*), which distinguishes it from photoperiodic suppression of NMDA- (*Figure 1—figure supplement 1*) and light-induced resetting (*Pittendrigh et al., 1984*; *vanderLeest et al., 2009*). Overall, these results confirm that the seasonal switch in GABA$_A$ signaling modulates the intrinsic responses of the SCN molecular clock (*DeWoskin et al., 2015*). Notably, the effect on waveform is

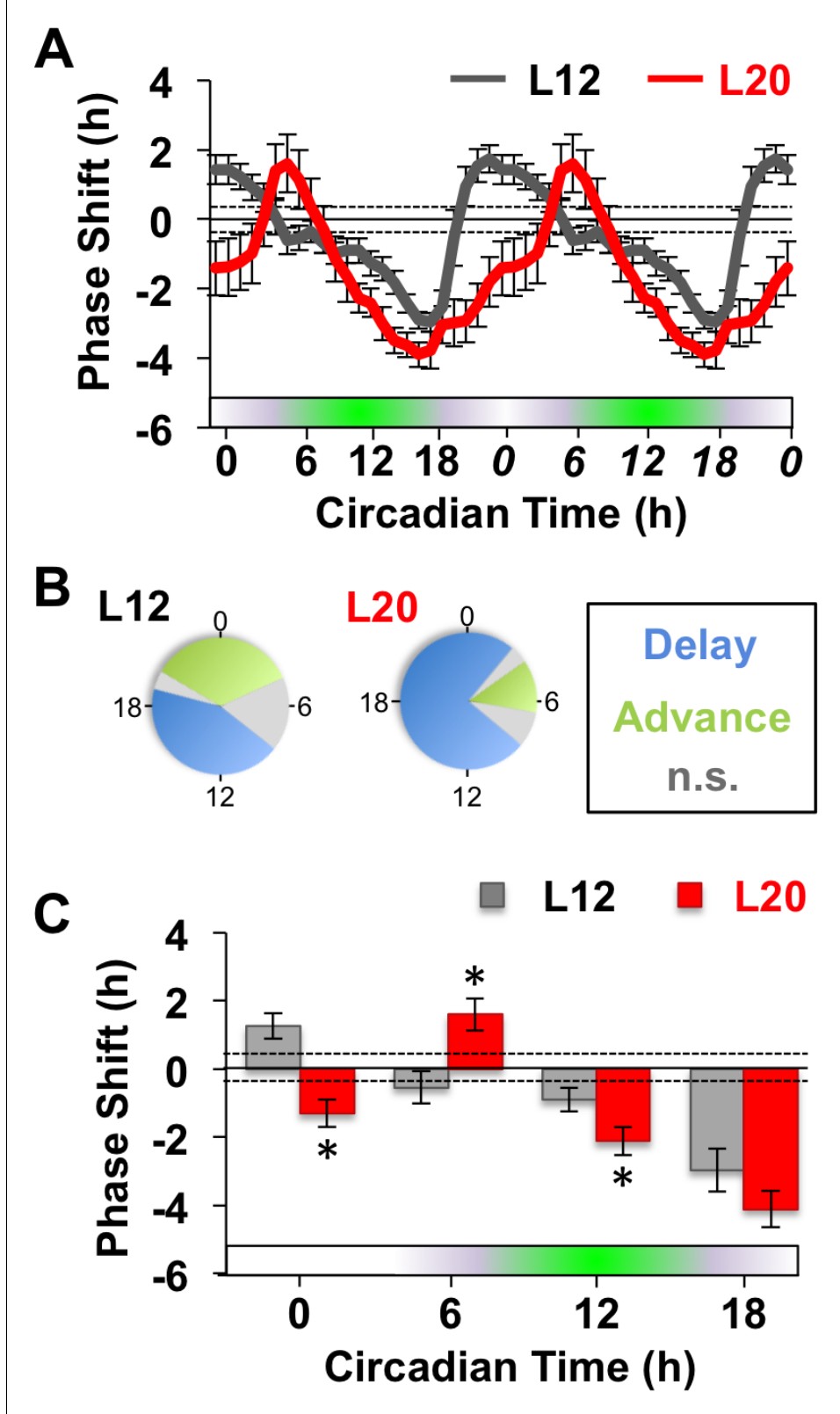

**Figure 1.** Long days alter SCN molecular responses to GABA. (**A**) The phase response curve to GABA is altered by photoperiodic history. Note that data are double plotted to illustrate responses spanning the circadian cycle, with double-plotted timepoints italicized. Stippled lines surrounding zero indicate mean response to vehicle pulses (see also *Figure 1—figure supplement 1*). Green and white bar along abscissa indicates the circadian time of the GABA pulse relative to the peak time of the PER2::LUC rhythm (green shading). For representative traces of PER2::LUC rhythms, please see
*Figure 1 continued on next page*

Figure 1 continued

**Figure 1—figure supplement 1.** (**B**) Polar plots illustrating that photoperiod alters the waveform of the GABA-resetting response rhythm. (**C**) Data binned for statistical analyses. n = 15–21 (CT0), 6–8 (CT6), 13–14 (CT12), and 6–8 (CT18) SCN slices/photoperiod. * Differs from L12 control, post-hoc Least Square Means contrasts, p<0.01. n.s.: phase with non-significant resetting relative to vehicle controls. See **Supplementary file 1** for full statistical results.

DOI: https://doi.org/10.7554/eLife.49578.003

The following figure supplement is available for figure 1:

**Figure supplement 1.** GABA- and NMDA-induced resetting of SCN PER2::LUC rhythms in vitro.

DOI: https://doi.org/10.7554/eLife.49578.004

of particular interest because this parameter often reflects changes in SCN organization (*Pittendrigh et al., 1984*; *vanderLeest et al., 2009*).

## Long days eliminate SCN rhythms in KCC2/NKCC1

Next we tested cellular mechanisms that could alter GABA responses of the SCN network. Long days increase mRNA encoding NKCC1 relative to KCC2 in the SCN (*Myung et al., 2015*), but the switch to excitatory GABA$_A$ signaling in other neural circuits of the adult brain is often driven by post-translational downregulation of KCC2 (*Chung, 2012*; *Hewitt et al., 2009*; *Lee et al., 2011*; *Ostroumov et al., 2016*; *Sarkar et al., 2011*). To test how day length modulates chloride co-transporter expression at the protein level, we measured KCC2 and NKCC1 in SCN slices collected from L12 and L20 mice at time points spanning the light:dark cycle. Based on previous reports documenting regional differences in GABA responses and chloride co-transporter expression in the rat SCN (*Belenky et al., 2010*; *Albus et al., 2005*), we analyzed KCC2 and NKCC1 separately in the two major subdivisions of the SCN network (i.e., shell and core; *Abrahamson and Moore, 2001*) using arginine vasopressin (AVP) expression to demarcate the SCN shell (*Figure 2A*, *Figure 2—figure supplement 1*). Lastly, we also used ratiometric analyses of KCC2/NKCC1 expression (*Figure 2B*) given that relative co-transporter expression influences intracellular chloride concentration (*Kaila et al., 2014*; *Lee et al., 2011*). Importantly, cellular patterns of KCC2 and NKCC1 immunoreactivity were consistent with previous studies (*Kaila et al., 2014*), with KCC2 immunoreactivity largely confined to the plasma membrane and NKCC1 immunoreactivity evident in both membrane and cytoplasm (*Figure 2—figure supplement 1*).

First we examined chloride co-transporter expression under L12 because there is scarce information on spatiotemporal patterns in the murine SCN. Under L12, KCC2/NKCC1 fluctuated over the day (*Figure 2B–C*) in the specific SCN compartment receiving dense retinal innervation (*Abrahamson and Moore, 2001*). Specifically, KCC2/NKCC1 varied in the SCN core (*Figure 2C*, Circwave cosinor rhythmicity test: p<0.05), but not in the SCN shell (*Figure 2C*, Circwave cosinor rhythmicity test: p>0.5) due to region-specific expression of both chloride co-transporters (*Figure 2D*, *Figure 2—figure supplement 1*). First for KCC2, expression was lower in AVP+ regions than in non-AVP-expressing regions of not only the SCN, but also the SON and PVN (*Figure 2—figure supplement 1*), consistent with work in the rat (*Belenky et al., 2010*; *Haam et al., 2012*). In the SCN, KCC2 expression was 91-fold lower in the AVP+ shell versus the non-AVP-expressing core (Student's t: p<0.005), but KCC2 levels did not vary over the day in either region (*Figure 2D*, Circwave cosinor rhythmicity test: p>0.2). In contrast, NKCC1 expression fluctuated over the day in both SCN compartments (*Figure 2D*, Circwave cosinor rhythmicity test: p<0.005 for each region), which is associated with daily KCC2/NKCC1 fluctuations in the SCN core (*Figure 2C*). These results establish that KCC2 and NKCC1 in the SCN are regulated in a spatiotemporal manner under standard lighting conditions, which may contribute to regional differences in clock neuronal excitability and GABA$_A$ responses (*Albers et al., 2017*; *Albus et al., 2005*; *Klett and Allen, 2017*; *Choi et al., 2008*).

Notably, L20 eliminated the daily rhythm of KCC2/NKCC1 in the SCN core (*Figure 2B–C*, Circwave cosinor rhythmicity test: p>0.4) by decreasing the mean daily expression of KCC2 (*Figure 2D–E*, LSM contrasts: p<0.01). Also in the SCN core, L20 attenuated the daily rhythm in NKCC1 expression (*Figure 2D*, Circwave cosinor rhythmicity test: p=0.1), but did not alter its levels across the day (*Figure 2F*, LSM contrasts: p>0.8) or at individual time points (*Figure 2D*, Full Factorial ANOVA: p=0.09). In the SCN shell, L20 increased NKCC1 expression (*Figure 2F*, LSM contrasts: p<0.01) and disrupted the daily rhythm of NKCC1 (Circwave cosinor rhythmicity test: p>0.1) due to elevation at

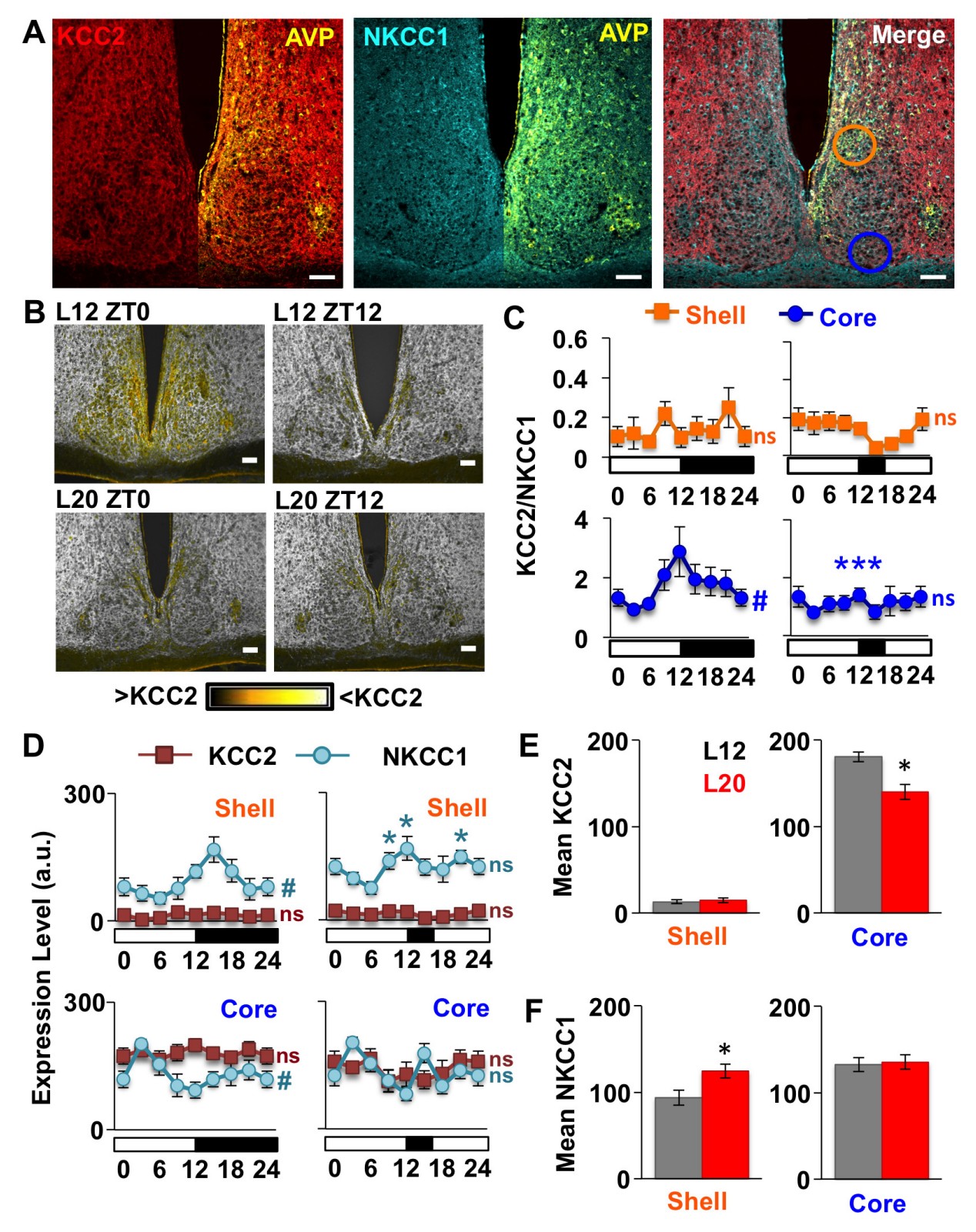

**Figure 2.** Long days eliminate region-specific rhythms in KCC2/NKCC1 expression. (A) Representative images illustrating AVP, KCC2, NKCC1 in the L12 SCN at the time of lights-off (Zeitgeber Time 12, ZT12). Blue and orange circles on merged image represent regions used for analyses of SCN shell and core, respectively. (B) Ratiometric images illustrate variation in KCC2/NKCC1 expression in L12 mice and loss of the KCC2/NKCC1 rhythm in L20 mice. Scalebar = 50 μm. (C) The daily expression of KCC2/NKCC1 is rhythmic in the SCN core under L12, but not under L20 (n = 4–5 mice/timepoint/

*Figure 2 continued on next page*

*Figure 2 continued*

photoperiod). (D) Photoperiodic modulation of KCC2 and NKCC1 expression in the SCN. (E-F) L20 reduces daily mean KCC2 expression in the SCN core and increases daily mean NKCC1 expression in the SCN shell. # significant rhythm, CircWave cosinor test, p<0.05, n.s. not rhythmic. * Differs from L12 control, post-hoc Least Square Means contrasts, p<0.01 (color-coded for chloride co-transporter in 2D). See *Supplementary file 1* for full statistical results.

DOI: https://doi.org/10.7554/eLife.49578.005

The following figure supplement is available for figure 2:

**Figure supplement 1.** Spatial, temporal, and photoperiodic changes in chloride co-transporter expression.

DOI: https://doi.org/10.7554/eLife.49578.006

specific times of the light:dark cycle (*Figure 2D*, LSM contrasts: p<0.01). This produced a slight decrease in overall KCC2/NKCC1 levels (L20: 14.83/124.61 = 0.12, L12: 13.16/93.97 = 0.14) but did not markedly alter daily expression of KCC2/NKCC1 in the SCN shell (*Figure 2C*). These results indicate that long days regulate the expression of both chloride co-transporters in the SCN, with decreased KCC2 in the SCN core, increased NKCC1 in the SCN shell, and elimination of the NKCC1 rhythm in both compartments. Given that small changes in chloride co-transporter expression can alter the electrochemical gradient of chloride (*Rivera et al., 2002*; *Woodin et al., 2003*), these collective changes would be predicted to increase the probability of depolarizing responses to GABA in each SCN compartment. This is consistent with previous work demonstrating that long days elevate intracellular chloride, depolarize the chloride equilibrium potential, and increase the number of SCN neurons that exhibit excitatory responses to GABA (*Myung et al., 2015*; *Farajnia et al., 2014*).

## Photoperiodic changes in NKCC1 and KCC2 function determine the dynamics of SCN coupling

To directly test the functional significance of photoperiodic changes in chloride co-transporter expression, we next examined whether inhibition of KCC2 or NKCC1 activity would modulate SCN network dynamics using selective antagonists of each chloride co-transporter (*Haam et al., 2012*; *Choi et al., 2008*). To test effects of chloride transport on network encoding, SCN slices were collected from L12 and L20 mice, then cultured with the NKCC1 antagonist bumetanide (BU, 80 μM), the KCC2 antagonist VU0240551 (VU, 80 μM), or vehicle-treated medium. In L12 slices, we found that both BU and VU increased SCN period by 0.35 hr (*Figure 3—figure supplement 1*), which was blocked by co-culture with the GABA$_A$ receptor antagonists bicuculline or gabazine (*Figure 3—figure supplement 1*). Importantly, these results indicate that inhibition of KCC2 and NKCC1 modulates the function of the SCN molecular clock by altering GABA$_A$ signaling, which is consistent with previous work demonstrating these compounds alter chloride transport, neuronal excitability, and GABA-evoked calcium responses (*Farajnia et al., 2014*; *Haam et al., 2012*; *Deisz et al., 2014*). At the cellular level, VU or BU increased the period of both SCN shell and core neurons in L12 slices (*Figure 3—figure supplement 1*), but did not markedly alter SCN organization (*Figure 3—figure supplement 1*) or damping of cellular rhythms (*Figure 3—figure supplement 1*). These data indicate that network and cell function was not adversely affected by drug treatment.

Consistent with previous work (*Evans et al., 2013*), in vivo exposure to L20 reorganized the SCN by imposing a large phase difference between the SCN shell and core on the first cycle in vitro (*Figure 3A*). This phase difference diminished over time in vehicle-treated slices (*Figure 3A*), which is driven by tetrodotoxin-sensitive signaling mechanisms (*Evans et al., 2013*). To quantify the dynamic process of network recovery, we measured changes in the phase relationship of SCN shell and core neurons over time in vitro (*Figure 3B*, see Materials and methods). As predicted for a coupled oscillator system (*Hansel et al., 1995*), the magnitude and direction of the cellular responses depended on the core-shell relationship at the start of the recording (*Figure 3B*). Specifically, the shell-core phase difference was reduced in the 'negative' direction when SCN core neurons phase-led SCN shell neurons by 2–6 hr (*Figure 3B*), whereas it was reduced in the opposite 'positive' direction when SCN core neurons phase-led SCN shell neurons by 8–16 hr (*Figure 3B*). Analysis of cellular period indicated that this coupling response is driven largely by changes in the cellular rhythms of SCN core neurons, which adopted a longer period when the network re-synchronized in the negative direction (*Figure 3C*, *Figure 4—figure supplement 1*) and a shorter period when re-synchronization occurred in the opposite direction (*Figure 3C*, *Figure 4—figure supplement 1*). Period modulation

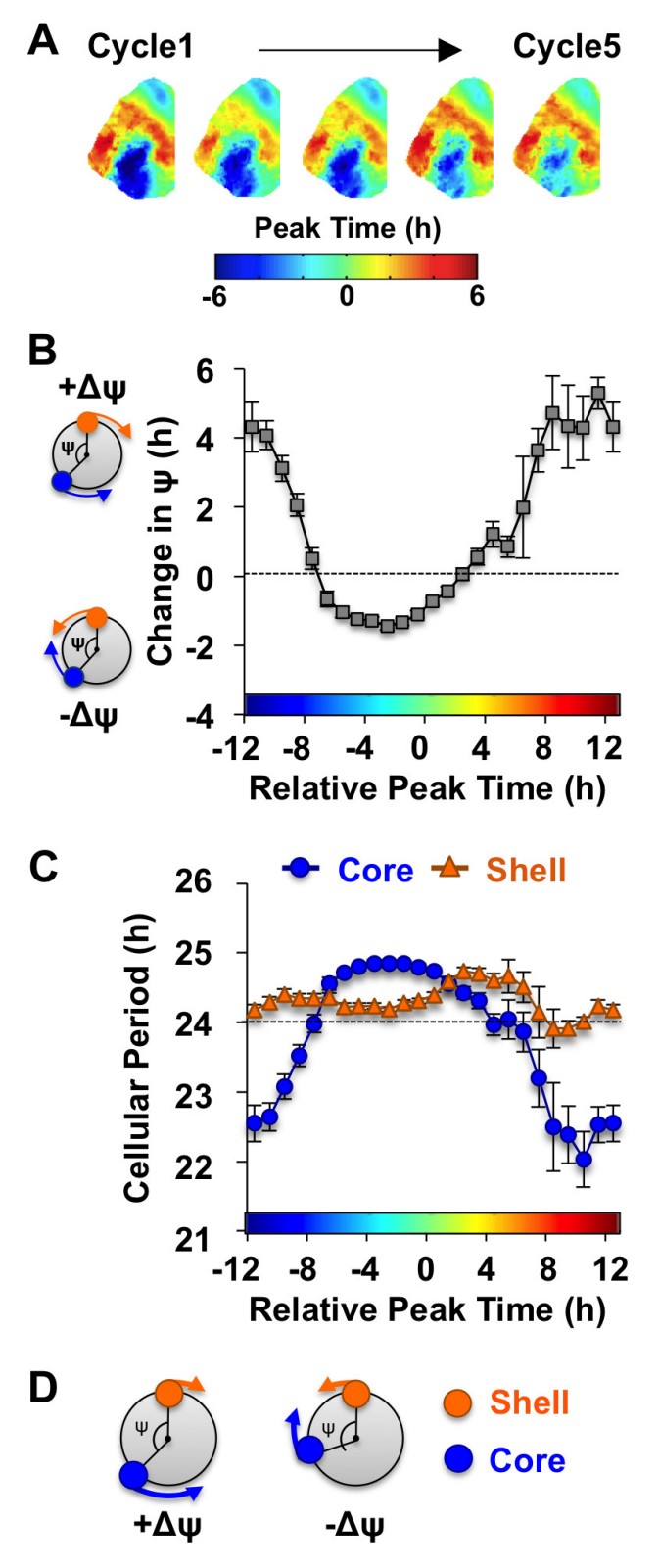

**Figure 3.** The SCN network recovers from the reorganized state induced by L20 due to intercellular signals that alter the phase and period of SCN core neurons. (**A**) Composite phase maps illustrating the reduction in the shell-core phase difference over the first five cycles in vitro in vehicle-treated SCN slices collected from L20 mice. (**B**) Coupling response in vehicle-treated SCN slices collected from L20 mice, as quantified by the change in the shell-core phase difference (Change in ψ) over time in vitro (n = 35 L20 slices, see *Figure 4—figure supplement 1* for cellular sample sizes). The color-coded
*Figure 3 continued on next page*

*Figure 3 continued*

bar along the abscissa represents the phase difference between shell and core neurons on Cycle 2 of recording (0 = core and shell neurons peak at same time, −8 = core neurons phase-lead by 8 hr). Polar plots along ordinate depict the direction of the coupling response for the positive and negative portion of the coupling response curve (blue and orange symbols represent core and shell neurons, respectively). (C) Cellular period responses of SCN core and shell neurons during network coupling. (D) Polar plots depicting cellular responses during network coupling in the positive and negative directions. See **Supplementary file 2** for statistical results.

DOI: https://doi.org/10.7554/eLife.49578.007

The following source data and figure supplement are available for figure 3:

**Source data 1.** Cellular data for coupling and period responses in **Figure 3** (Vehicle conditions).
DOI: https://doi.org/10.7554/eLife.49578.009
**Figure supplement 1.** Modulation of chloride co-transporter function influences L12 SCN period.
DOI: https://doi.org/10.7554/eLife.49578.008

in SCN shell neurons was less affected by phase relationship (*Figure 3C*, *Figure 4—figure supplement 1*), suggesting that this specific neuronal subpopulation did not shift much under vehicle conditions. Overall, these results indicate that the SCN network recovers from its reorganized state in vitro largely due to intercellular signals that modulate the phase and period of SCN core neurons (*Figure 3D*).

Notably, the dynamics of this coupling response were modulated by KCC2 and NKCC1 inhibition, with complementary changes depending on which chloride co-transporter was targeted (*Figure 4A*, *Figure 4—figure supplement 1*). When L20 SCN slices were cultured with the KCC2 inhibitor VU, the process of network resynchronization was accelerated in the negative direction (*Figure 4A–B*, *Figure 4—figure supplement 1*) due to altered period responses of SCN core neurons (*Figure 4C–D*, *Figure 4—figure supplement 1*). In contrast, BU inhibition of NKCC1 slowed network recovery (*Figure 4A–B*, *Figure 4—figure supplement 1*) by affecting the period of both SCN shell and core neurons (*Figure 4C–D*, *Figure 4—figure supplement 1*). Interestingly, BU blocked the period responses of SCN core neurons markedly when the network was reorganized by L20 (*Figure 4—figure supplement 1*), which is consistent with the decreased KCC2 in this region under this photoperiod (*Figure 2D*). Collectively, these data suggest that photoperiodic changes in KCC2/NKCC1 activity determine the rate and dynamics of network recovery by modulating $GABA_A$ signaling in the SCN core.

To further test the role of decreased KCC2 in modulating network coupling after L20, we treated SCN slices with the KCC2 activator CLP290 (*Gagnon et al., 2013*). CLP290 is a carbamate prodrug that increases KCC2 surface expression, reduces intracellular chloride, and restores $E_{GABA}$ in mature neurons with diminished KCC2 expression (*Ostroumov et al., 2016*; *Gagnon et al., 2013*; *Ferrini et al., 2017*; *Chen et al., 2017*). Based on the ratiometric model of chloride flux, we predicted that enhancing KCC2 function would mimic the results of the NKCC1 antagonist and likewise slow SCN recovery. To test whether restoration of KCC2 function would adversely affect SCN coupling, we treated L12 and L20 slices with 100 μM CLP290, as in *Chen et al. (2017)*. Similar to VU and BU, CLP290 increased the period of L12 SCN slices by lengthening period in both shell and core neurons (*Figure 3—figure supplement 1*), but did not markedly alter network organization (*Figure 3—figure supplement 1*) or cellular damping over time in vitro (*Figure 3—figure supplement 1*). Consistent with predictions, CLP290 attenuated network recovery in L20 slices (*Figure 4A–B*) due to abrogation of coupling responses in both SCN core and shell neurons (*Figure 4C–D*, *Figure 3—figure supplement 1*). Overall, the effects of CLP290 in L20 slices were qualitatively similar to those induced by BU (*Figure 4E*), providing complementary evidence that the photoperiodic decrease in KCC2-mediated chloride transport is a critical modulator of intercellular coupling during network recovery from the long day state.

## NKCC1 modulates network recovery from long days in vivo

Given that BU delayed network recovery in vitro, we next tested whether it would likewise modulate circadian behavior in vivo. Bumetanide is an FDA-approved diuretic used in humans and rodent models, which has been shown to influence neural function when administered orally (*Tyzio et al., 2014*). To test the effects of BU on circadian behavior, L12 and L20 mice were singly housed in wheel-running cages and provided BU (5 mg/kg) or vehicle in the drinking water. After

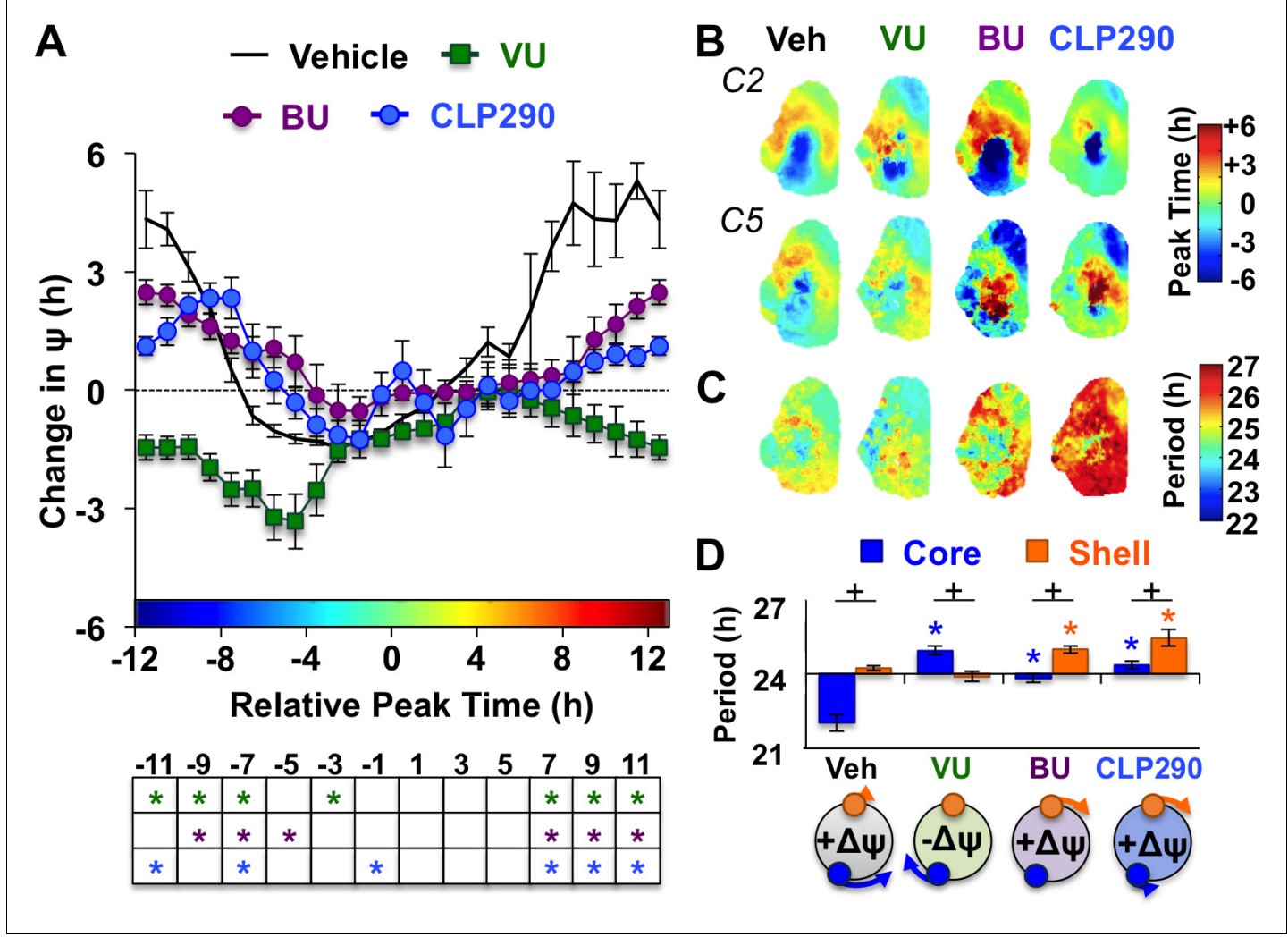

**Figure 4.** KCC2 and NKCC1 influence network dynamics during SCN coupling. (**A**) Coupling response curves for Vehicle, VU, BU, and CLP290 treatment (n = 6 L20 slices/antagonist, see *Figure 4—figure supplement 1* for cellular sample sizes). (**B**) Average phase maps depicting SCN spatiotemporal organization on Cycle 2 (C2) and Cycle 5 (C5) in vitro illustrating drug-induced effects on the magnitude and direction of network coupling. (**C**) Average period maps illustrating region-specific effects on cellular period. (**D**) Average period length (± SEM) of SCN shell and core neurons provided Vehicle, VU, BU, or CLP290 treatment while in the reorganized state induced by L20 (i.e., core-shell relationship = −11 hr ± 2 hr). See *Figure 4—figure supplement 1* for full period response curves. Below X-axis: Schematic illustrating how network coupling is altered by region-specific modulation of period responses, with the net effect on the direction of network resynchronization indicated within each polar plot. * Differs from Vehicle control, post-hoc Least Square Means contrasts, p<0.01 (color-coded by antagonist in 4A), + Differs from within-group complementary region, LSM contrasts, p<0.05. See *Supplementary file 2* for full statistical results. Other conventions as in *Figure 3*.
DOI: https://doi.org/10.7554/eLife.49578.010

The following source data and figure supplement are available for figure 4:

**Source data 1.** Cellular data for coupling and period responses in *Figure 4* (BU, VU, and CLP290 antagonist conditions).
DOI: https://doi.org/10.7554/eLife.49578.012
**Figure supplement 1.** Full coupling and period response curves under Vehicle (**A**), VU (**B**) and BU (**C**) and CLP290 (**D**) conditions (L20: n = 8–35 slices/group).
DOI: https://doi.org/10.7554/eLife.49578.011

administration in drinking water, BU concentration reached 5 µM in plasma and 40 nM in the hypothalamus, the latter of which approaches the half-maximal inhibitory concentration of BU for NKCC1 (*Russell, 2000*). After 8 weeks of photoperiodic pre-treatment with or without BU, L12 and L20 mice were released into constant darkness (DD, *Figure 5A*, *Figure 5—figure supplement 1*) to monitor changes in circadian behavior reflective of SCN coupling (*Evans and Gorman, 2016*;

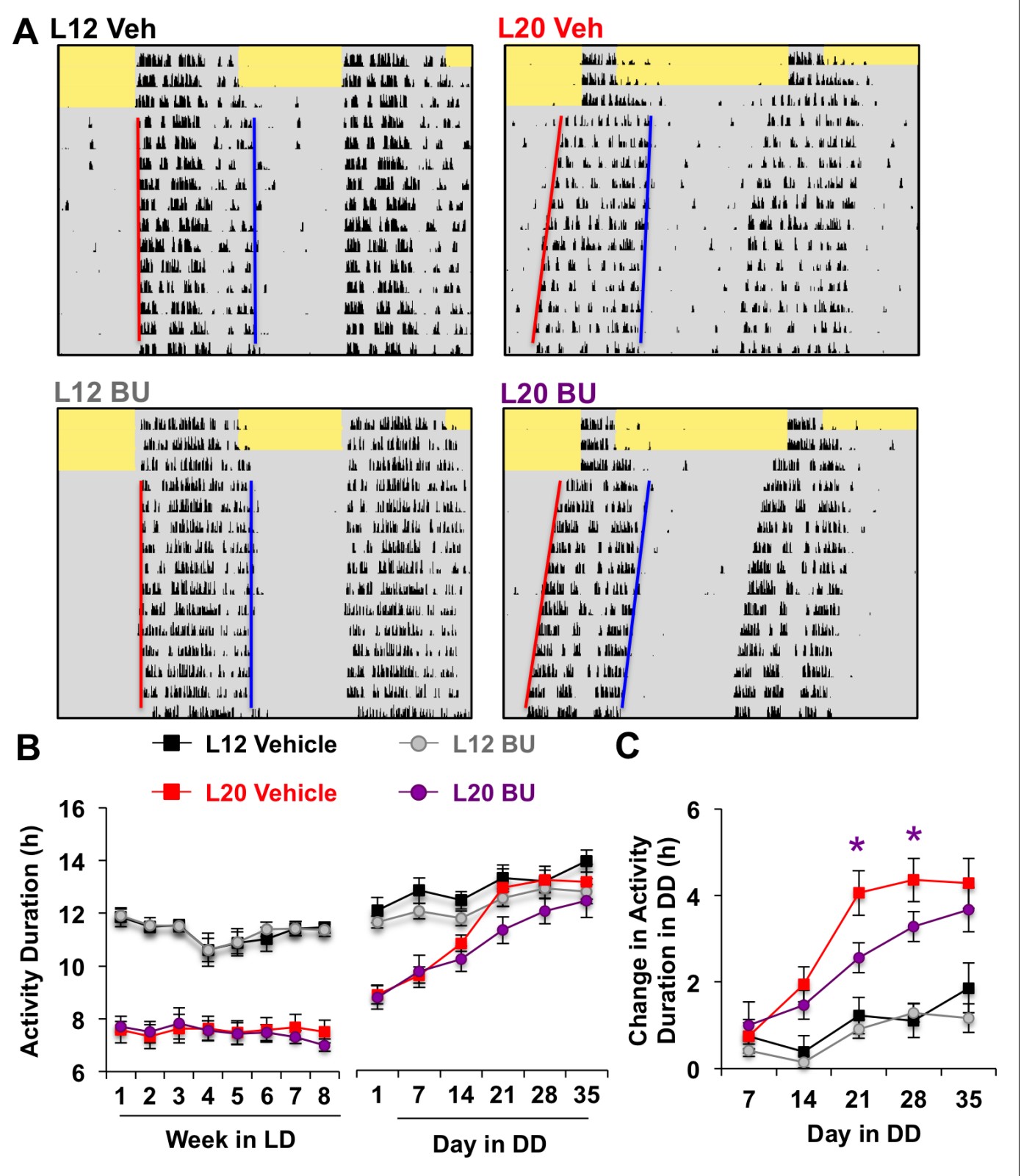

**Figure 5.** Photoperiodic changes in NKCC1 modulate circadian plasticity in vivo after release into constant darkness (DD). (**A**) Double-plotted actograms representing effects of bumetanide (BU) on circadian waveform during entrained and DD conditions. Yellow and gray shading represent light and darkness, respectively. Superimposed red and blue lines illustrate free-running period of activity onset and offset, respectively. See *Figure 5—figure supplement 1* for additional representative actograms. (**B**) Photoperiod, but not BU, influenced activity duration during photo-entrainment

*Figure 5 continued on next page*

Figure 5 continued

(n = 6 mice/photoperiod/treatment). (C) BU influenced plasticity in circadian waveform after release from L20 into DD by slowing the expansion of activity duration. * Differs from photoperiod-matched vehicle control, LSM contrasts, p<0.05. See *Supplementary file 1* for full statistical results.

DOI: https://doi.org/10.7554/eLife.49578.013

The following figure supplement is available for figure 5:

**Figure supplement 1.** Effects of systemic BU administration in vivo.

DOI: https://doi.org/10.7554/eLife.49578.014

*Pittendrigh and Daan, 1976a*). Specifically, exposure to long days alters circadian waveform in nocturnal rodents by compressing the duration of the active phase, which is confirmed by decompression after release into DD. Importantly, the waveform of daily rhythms is closely associated with changes in SCN state, and decompression of the active phase manifests as the SCN returns to its synchronous state (*Evans et al., 2013*; *Margraf et al., 1991*; *Jagota et al., 2000*). Long days also shorten free-running period, but this aftereffect persists after SCN organization has recovered in vivo (*Evans et al., 2013*; *Myung et al., 2015*; *Pittendrigh and Daan, 1976b*).

Consistent with results obtained in vitro, BU slowed the recovery of circadian waveform after release from L20 into DD in vivo (*Figure 5A–C*, *Figure 5—figure supplement 1*). L20 mice treated with BU displayed slower recovery of circadian waveform over time in DD due to decreased rate of activity decompression (Repeated Measures ANOVA: Drug*Time p<0.05), but BU did not significantly alter behavior after release from L12 (Repeated Measures ANOVA: Drug*Time p>0.5). Further, BU did not alter wheel running rhythms under entrained conditions, (*Figure 5B*, Repeated Measures ANOVA: p>0.5), SCN organization, body weight, water intake, activity levels, free-running period, phase angle of entrainment, or behavioral adjustment to simulated jetlag (*Figure 5—figure supplement 1*). This pattern of results suggests that photoperiodic changes in the relative role of NKCC1 regulate the rate at which behavior recovers from exposure to long days.

## Excitatory GABA$_A$ signaling is required for restoration of SCN phase synchrony

Our results indicate that exposure to long days modulates chloride transport in the SCN by decreasing KCC2 in the specific SCN region that processes light input. Because downregulation of KCC2 alters the sign of GABA$_A$ signaling in other regions of the adult brain (*Hewitt et al., 2009*; *Lee et al., 2011*; *Ostroumov et al., 2016*; *Sarkar et al., 2011*), we hypothesized that non-canonical GABA responses (i.e., depolarization) modulate the dynamics of SCN coupling after exposure to long days. Depolarizing responses to GABA are driven by anionic gradient shifts in the flux of both chloride and bicarbonate (*Staley et al., 1995*; *Rivera et al., 2005*), and bicarbonate regeneration is necessary for excitatory GABA$_A$ responses (*Ostroumov et al., 2016*; *Staley et al., 1995*). Thus, we predicted that if SCN recovery from L20 requires excitatory GABA$_A$ responses, then network resynchronization would be attenuated when bicarbonate regeneration is inhibited by the carbonic anhydrase antagonist, acetazolamide (*Ostroumov et al., 2016*; *Staley et al., 1995*). Mice refused to drink water treated with acetazolamide, thus we returned to our in vitro assay to track SCN coupling in real time.

SCN slices from L12 and L20 mice were cultured with either acetazolamide (ACE, 100 µM) or vehicle (<0.1% DMSO). Similar to VU, BU, and CLP290, ACE increased L12 SCN period in a GABA$_A$-dependent manner (*Figure 6—figure supplement 1*). ACE also eliminated the period difference between SCN core and shell neurons in L12 SCN slices by specifically increasing the period of SCN shell neurons (*Figure 6—figure supplement 1*), consistent with low KCC2/NKCC1 expression in this region (*Figure 2C*). Interestingly, equalization of cellular period reduced the shell-core phase difference that typically manifests in L12 slices over time in vitro (*Figure 6—figure supplement 1*) or after release into DD (*Evans et al., 2013*; *Evans et al., 2011*). This suggests that excitatory GABA$_A$ signaling in SCN shell neurons imposes the shell-core phase difference, consistent with previous work using the non-selective chloride transport inhibitor furosemide (*Myung et al., 2015*).

Because L20 decreased KCC2 in the SCN core, we predicted that ACE would exert more pronounced effects under this condition. Remarkably, ACE treatment locked the SCN network in the reorganized L20 state (*Figure 6A–B*) by specifically blocking the period response of SCN core

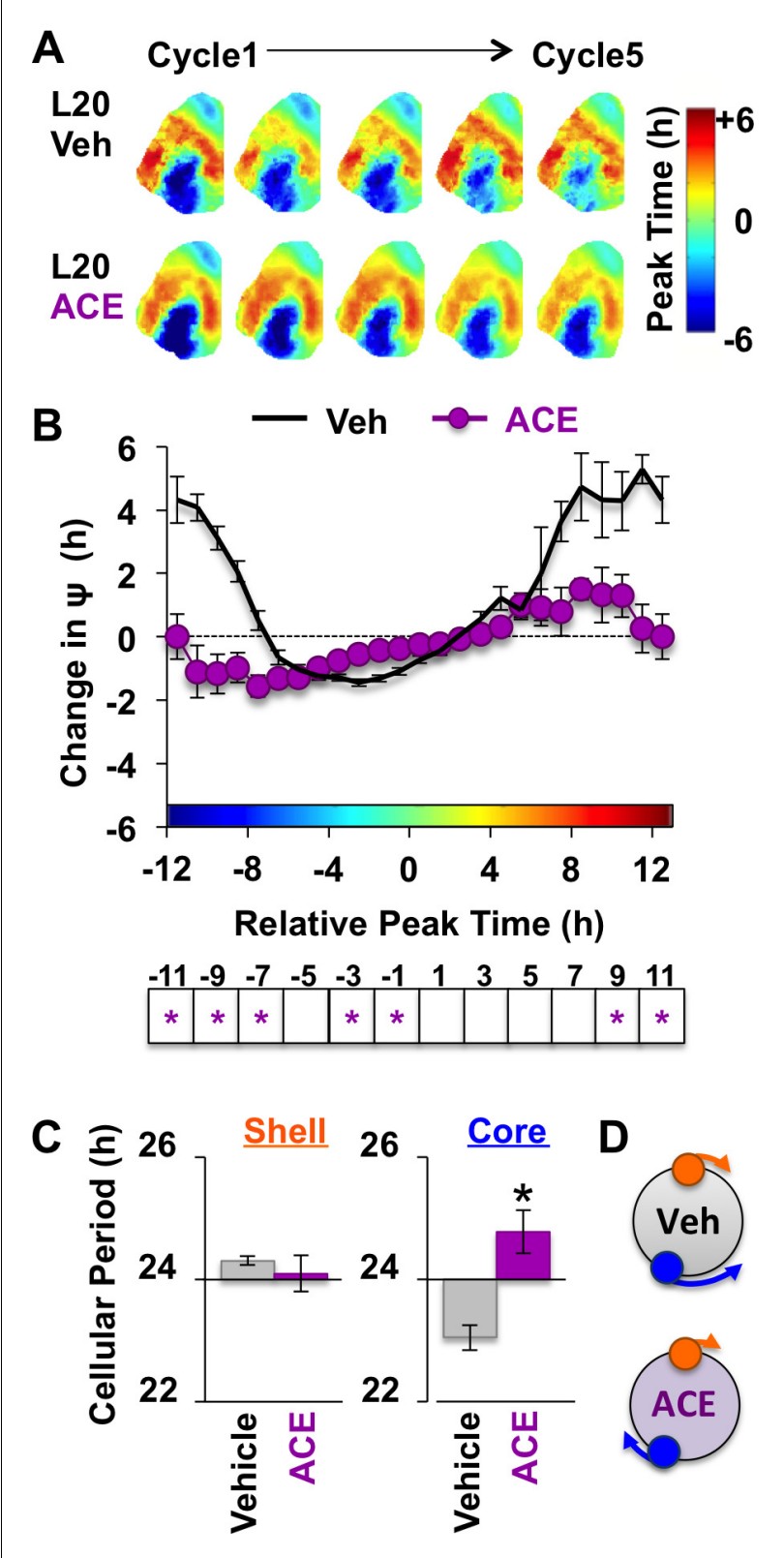

**Figure 6.** Non-canonical GABA$_A$ signaling is necessary for restoring phase synchrony in the SCN network in vitro. (**A**) Average phase maps for L20 slices treated with ACE or vehicle-treated medium. (**B**) ACE attenuates SCN coupling responses after network reorganization by L20 (ACE n = 16 L20 slices, see *Figure 6—figure supplement 1* for cellular sample sizes). (**C**) ACE lengthens the period response of SCN core neurons after network reorganization by L20 (i.e., core-shell relationship = −11 hr ± 2 hr, see *Figure 6—figure supplement 1* for full period response curves). (**D**) Schematic

*Figure 6 continued on next page*

*Figure 6 continued*

illustrating effects of ACE on cellular period and coupling response in the polarized SCN state induced by L20. * Differs from vehicle control, LSM contrasts, p<0.05. See *Supplementary file 2* for full statistical results. Other conventions as in *Figure 4*.

DOI: https://doi.org/10.7554/eLife.49578.015

The following source data and figure supplement are available for figure 6:

**Source data 1.** Cellular data for coupling and period responses in *Figure 6* (ACE conditions).

DOI: https://doi.org/10.7554/eLife.49578.017

**Figure supplement 1.** Non-canonical GABA$_A$ signaling modulates L12 SCN by affecting the period of SCN shell neurons.

DOI: https://doi.org/10.7554/eLife.49578.016

neurons (*Figure 6C–D*, *Figure 6—figure supplement 1*). This is of interest because L20 decreased KCC2 expression (*Figure 2E*) and increased BU responsiveness (*Figure 4—figure supplement 1*) in this specific SCN compartment. Similar to effects produced by chloride transport inhibition, effects of ACE were region-specific and state-dependent because the coupling responses of SCN shell neurons were not altered when the network was in the reorganized state (*Figure 6—figure supplement 1*). Collectively, these data reveal that the photoperiodic switch in GABA$_A$ signaling is necessary for the functional restoration of the SCN network and that responses of SCN core neurons are those most influenced by the photoperiodic changes in GABA$_A$ signaling.

## Discussion

Light is the most salient cue for the SCN network, and seasonal changes in photic conditions are encoded via the spatiotemporal relationships of SCN cellular clocks (*Meijer et al., 2010*; *Evans and Gorman, 2016*). Given the ever-changing nature of the photic environment, there is a premium on maintaining plasticity of SCN encoding. Here we provide novel insight into seasonal encoding by revealing that the photoperiodic switch in GABA$_A$ signaling is necessary for restoring SCN function after it is disrupted by light. By examining dynamic changes in cellular synchrony, we have uncovered that plasticity in GABA$_A$ signaling is a critical neuroadaptation that modifies how SCN clock neurons communicate with one another to regulate circuit function in accordance with an ever-changing environment. This switch corresponds with KCC2 downregulation in the specific SCN compartment that receives and processes light input, which is the same region in L20 slices displaying altered cellular responses during pharmacological modulation of chloride transport and non-canonical GABA$_A$ signaling. Collectively, these results indicate that downregulation of KCC2 in this specific region is critical for network recovery after exposure to long days, and that this process is likely driven by mechanisms similar to those regulating GABA$_A$ signaling in other important networks controlling behavior.

Here we find that seasonal changes in GABA$_A$ signaling in the SCN network regulate the transition back to its synchronous state. In this manner, seasonal plasticity in GABA$_A$ signaling may be viewed as a homeostatic mechanism because it acts to restore circuit function after it is perturbed by light. Homeostasis is not a term commonly employed in a circadian context, but its applicability to seasonal encoding is not without precedent. For instance, formal assays have shown that photoperiodic adjustments in behavior and physiology return to steady-state when the long day stimulus is removed in vivo (*Pittendrigh and Daan, 1976a*). Further, photoperiodic alterations in SCN cellular relationships return to steady-state after DD release in vivo and culture ex vivo (*Evans et al., 2013*). Homeostatic recalibration is critical for most biological systems, but the underlying mechanisms remain poorly understood in many neural networks because circuit transitions are difficult to study in real-time. By tracking this dynamic process ex vivo, we first demonstrated an unexpected role for GABA$_A$ signaling in network recovery (*Evans et al., 2013*). Here we show this form of neuroplasticity requires non-canonical GABA$_A$ signaling by demonstrating that ACE locks the SCN network into its reorganized state by modulating the coupling response of SCN core neurons. Interestingly, our data suggest that non-canonical GABA$_A$ signaling also modulates SCN dynamics under steady state by acting on a different cellular target. Specifically, ACE influenced cellular relationships in L12 slices due to modulation of the molecular clock in SCN shell neurons. This equalized cellular period across the network and attenuated the shell-core phase difference that manifests over time in culture or after release into DD in vivo (*Evans et al., 2013*; *Evans et al., 2011*). This indicates that non-

canonical GABA$_A$ signaling acts to impose phase differences during steady state by modulating the period of SCN shell neurons, but facilitates synchrony in the polarized state by acting on SCN core neurons. Recent reports indicate that GABA$_A$ signaling is involved in other types of network behaviors not examined directly here (*Freeman et al., 2013*; *Azzi et al., 2017*), thus it would be of interest to determine whether non-canonical responses to GABA are likewise involved in these other SCN processes. Further work investigating how state-dependent GABA signaling regulates cellular physiology in different subclasses of SCN neurons is warranted given that this is the major neurotransmitter expressed by SCN neurons (*Albers et al., 2017*).

In addition to providing novel insight into the functional significance of seasonal changes in GABA$_A$ signaling, the present results expand understanding of the cellular mechanisms underlying this form of neuroplasticity. Neuronal responses to GABA$_A$ signaling are influenced by the ratio of KCC2 and NKCC1 expression, and changes in either co-transporter can invert ion flux because the equilibrium potential for chloride is close to the resting membrane potential (*Rivera et al., 2002*; *Woodin et al., 2003*). In other adult networks, KCC2 downregulation is the key event modulating the sign of GABA$_A$ signaling because it alters the co-transporter ratio to favor NKCC1-mediated Cl$^-$ influx (*Hewitt et al., 2009*; *Lee et al., 2011*; *Ostroumov et al., 2016*; *Sarkar et al., 2011*). Here we show that long days decrease KCC2 in the SCN core and increase NKCC1 in the SCN shell, consistent with previous work demonstrating that long days increase depolarizing responses to GABA in both SCN compartments (*Farajnia et al., 2014*). Given the limits of drawing inferences from expression alone, we next tested the functional implications of photoperiodic changes in KCC2/NKCC1 using two different approaches (i.e., NKCC1 antagonism with bumetanide, KCC2 agonism with CLP290). Both drugs attenuated the ability of the long-day SCN network to return to its basal state ex vivo in a qualitatively similar manner, suggesting that photoperiodic adjustments of chloride transport serve to speed SCN recovery after disruption by light. Notably, long days decreased KCC2 in the SCN core during the daytime, which matches the phase when SCN neuronal responses to GABA are affected most by photoperiod (*Farajnia et al., 2014*). Further, KCC2 antagonism accelerated SCN recovery in vitro by changing the coupling responses of SCN core neurons, suggesting that further suppression of KCC2 activity beyond that achieved by light may have potential benefits. Lastly, we find that ACE blocked network recovery by specifically modulating the coupling responses of SCN core neurons. The convergent nature of these results is consistent with the known effects of these compounds on chloride transport, neuronal excitability, and GABA-evoked calcium responses (*Farajnia et al., 2014*; *Haam et al., 2012*; *Deisz et al., 2014*). Collectively, these results indicate that KCC2 downregulation in the SCN core is the critical cellular adaptation driving network recovery in the master clock. It remains unclear how light modulates KCC2 expression, but it is likely that this photoperiodic change is achieved via post-translational regulation of protein expression and/or activity because long days do not increase *Kcc2* transcription (*Myung et al., 2015*). Indeed, KCC2 function can be controlled by its phosphorylation state (*Kaila et al., 2014*), which is modulated by intercellular signaling mechanisms known to be involved in photic processing, including glutamate, BDNF, and extrasynaptic GABA signaling (*Albers et al., 2017*). Given that KCC2 is downregulated in the SCN compartment that receives and processes photic input, identifying the specific pathways by which light represses KCC2 in this region represents an important area for future work.

Consistent with its effects in vitro, we found that bumetanide reduced circadian plasticity in vivo. Although the systemic approach used here did not target a specific locus, bumetanide specifically decreased plasticity of circadian waveform, which is a rhythmic property closely linked to spatiotemporal encoding by the SCN network (*Evans and Gorman, 2016*; *Pittendrigh and Daan, 1976a*; *Margraf et al., 1991*; *VanderLeest et al., 2007*). Slower recovery of behavioral state following systemic bumetanide administration in vivo is similar to the reduced rate of SCN recovery in vitro. In contrast, in vivo bumetanide administration did not prevent long day encoding itself or the aftereffect on free-running period, which may reflect that brain levels of BU were insufficient to block these effects. Hypothalamic levels of bumetanide in the current study approached the half-maximal inhibitory concentration for NKCC1 (*Russell, 2000*; *Löscher et al., 2013*; *Hampel et al., 2018*). Systemically administered bumetanide has been shown to modulate hippocampal synaptic plasticity and memory in mouse models of disease (*Marguet et al., 2015*; *Deidda et al., 2015*), and the efficacy of this compound in previous and current work may relate to its prolonged administration over many days to mice, a species in which the plasma half-life of bumetanide is >45 min (*Töpfer et al., 2014*). Given that the SCN regulates both circadian period and waveform, future work targeting this

specific locus may provide additional insight into the processes controlling seasonal encoding and aftereffects. Although pharmacological manipulations may be limited by the need for continued drug efficacy over weeks without daily interference, the current results may be used to develop and optimize genetic approaches to modulate chloride transport and/or GABA$_A$ signaling specifically in SCN neurons. The current results indicate that cell identity and phasing may be important considerations in the design of these future studies.

In agreement with previous work suggesting daily SCN changes in GABA$_A$ signaling correspond with altered NKCC1 expression (*Belenky et al., 2010*; *Choi et al., 2008*), our results demonstrate that NKCC1 and KCC2/NKCC1 are regulated over the day under a standard lighting condition. Thus, the adult SCN is likely a network in which neuroplasticity of GABA$_A$ signaling influences circuit function on both a seasonal and circadian basis (*Albers et al., 2017*). The regional differences in KCC2/NKCC1 expression described here reflect disparities in both KCC2 and NKCC1 expression across SCN compartments. Similar to other AVP+ structures in the hypothalamus (*Haam et al., 2012*), we detected stark regional differences in KCC2 expression across the SCN shell and core. Further, the timing of peak NKCC1 expression appeared to differ across SCN compartment, consistent with Western blot analyses of NKCC1 in the mouse SCN conducted at two times of day (*Choi et al., 2008*). Interestingly, low NKCC1 expression in the SCN core corresponds with the phase of low intracellular chloride in this specific region (*DeWoskin et al., 2015*). Thus, daily variation in chloride co-transporter expression may contribute to documented regional differences in neuronal excitability and GABA$_A$ responses of SCN neurons (*Albers et al., 2017*; *Albus et al., 2005*; *Klett and Allen, 2017*; *Choi et al., 2008*). Considering that changes in KCC2 and NKCC1 can impact myriad other cellular processes (*Kaila et al., 2014*), further investigation into their role in the SCN network is warranted.

Seasonal changes in the SCN circuit also alter molecular responses to GABA stimulation. Phase resetting responses elicited here are consistent with findings that tonic GABA stimulation effectively shifts the phase of the SCN molecular clock (*DeWoskin et al., 2015*). One caveat to longer-term treatment is the inadvertent modulation of additional targets. With this in mind, it is notable that present responses to tonic GABA treatment appear similar in several respects to those elicited by more acute treatment (*Liu and Reppert, 2000*). For instance, GABA-induced resetting was occluded by GABA$_A$ antagonism, replicating previous work that also directly excluded a contribution of GABA$_B$ signaling (*Liu and Reppert, 2000*). In addition, it has been found that 1 hr and 6 hr pulses of GABA elicit phase shifts of similar magnitude (*Liu and Reppert, 2000*), suggesting that SCN resetting responses are not fundamentally altered by longer-term treatment. Consistent with this idea, the waveform of GABA-induced resetting displayed by L12 SCN slices replicates that elicited in previous work (*Liu and Reppert, 2000*). Although the absolute magnitude and precise phasing of resetting responses vary across studies, this may reflect other important methodological differences (e.g., phase via PER2::LUC versus electrical rhythms [*Liu and Reppert, 2000*], SCN slice from adult mouse versus dissociated SCN neurons from early postnatal pups [*Liu and Reppert, 2000*]). Going forward, the assay developed here may be useful to examine whether photoperiod modulates sensitivity to GABA signaling across the circadian cycle. Further, imaging may add insight into cell-type specific responses since the signal recorded with luminometry is a population-level rhythm that largely reflects the brightest regions of the slice (i.e., shell) (*Azzi et al., 2017*). The results of further studies such as these may provide additional insight into seasonal plasticity of GABA circuits and how these intersect with the intracellular circadian clock.

Last, the current results may have implications for annual changes in human health and physiology. Seasonal disorders manifest in 3–10% of people, with the greatest incidence occurring at high latitudes where seasonal variation in photoperiod is most pronounced (*Wirz-Justice, 2018*). Evidence supports a light-based, circadian basis for seasonality (*Wehr et al., 2001*; *Wirz-Justice, 2018*), but the neurobiological basis of human variation in sensitivity to seasonal disorders remains unknown. Like humans, many rodent species are characterized by two seasonal morphs: short day responders and short day non-responders (*Nelson, 1987*). The Siberian hamster, a classic model for interrogating photoperiodic circuits, typically displays seasonal alterations in reproduction, metabolism, immune function, affective state, and cognition (*Goldman, 1999*). However, a subset of Siberian hamsters fail to transition to the winter phenotype after exposure to long days due to a genetic pre-disposition that locks their SCN into a summer state (*Margraf et al., 1991*; *Puchalski and Lynch, 1991*; *Goldman et al., 2000*). It has been theorized that short day non-responsiveness

reflects individual differences in SCN coupling that hinders network plasticity. Here we demonstrate that $GABA_A$ signaling regulates seasonal plasticity in the state of the SCN network, raising the possibility that individual differences in SCN GABA circuits may contribute to natural variation in seasonal responsiveness. Thus, further investigation of mechanisms and factors that regulate photoperiodic remodeling of GABA circuits in the SCN may help to better understand the neural basis of annually reoccurring disease.

# Materials and methods

## Key resources table

| Reagent type (species) or resource | Designation | Source or reference | Identifiers | Additional information |
|---|---|---|---|---|
| Genetic reagent (*M. musculus*) | mPer2$^{Luc}$ mouse | *Yoo et al., 2004* | RRID: IMSR_JAX:006852 | |
| Antibody | anti-AVP (Guinea Pig polyclonal) | Peninsula Laboratories Cat# T-5048 | RRID: AB_2313978 | Primary antibody, (1:1K) |
| Antibody | anti-NKCC1 (Goat polyclonal) | Abcam Cat#ab99558 | RRID:AB_10675276 | Primary antibody, (1:500) |
| Antibody | anti-KCC2 (Rabbit polyclonal) | Millipore Cat# 07–432 | RRID:AB_310611 | Primary antibody, (1:500) |
| Antibody | Anti-Guinea Pig, Alexa Fluor 647 | Jackson Immunoresearch Cat#706-605-148 | RRID:AB_10895029 | Secondary antibody, (1:200) |
| Antibody | Donkey Anti-Goat, Alexa Fluor 555 | Abcam Cat#ab150130 | RRID:AB_10894526 | Secondary antibody, (1:200) |
| Antibody | Donkey Anti-Rabbit, Alexa Fluor 488 | Jackson Immunoresearch Cat#711-545-152 | RRID:AB_10893040 | Secondary antibody, (1:200) |
| Commercial assay or kit | Bumetanide ELISA Kit | Neogen Corporation Cat#103719–1 | | |

## Mice and husbandry conditions

All procedures involving mice were conducted according to the NIH Guide for the Care and Use of Animals and were approved by the Institutional Animal Care and Use Committees at Marquette University. Homozygous mPer2$^{Luc}$ mice (*Yoo et al., 2004*) were bred and raised under a standard 24 hr light:dark cycle with 12 hr light and 12 hr darkness (LD12:12, lights-off: 1800 CST). At 10–12 weeks of age, male mPer2$^{Luc}$ mice were individually housed with either 12 hr of light per day (L12, lights-off 1800 CST) or 20 hr of light per day (L20, lights-off: 1800 CST) for a minimum of 8 weeks. Ambient room temperature was maintained at 22 C $\pm$ 2 C under both colony and experimental conditions, and mice had *ab libitum* access to water and food (Teklad Rodent Diet #8604).

## SCN collection and ex vivo phase resetting assays

Coronal SCN slices (150 μm) were collected from mPer2$^{Luc}$ mice and cultured as described previously (*Evans et al., 2013*). Briefly, SCN slices were collected 4–6 hr before lights off, since dissections during late subjective day do not markedly reset the phase of the SCN (*Davidson et al., 2009*). Each SCN was cultured on a membrane insert in a dish containing 1.2 mL of air-buffered Dulbecco's modified explant medium (DMEM, Sigma D2902) supplemented with 0.1 mM beetle luciferin, 0.02% B27 (Gibco 17504), 0.01% HEPES (Gibco 15630), 0.005% $NaCHO_3$ (Gibco 25080), 0.004% Dextrose (Sigma G7021), and 0.01% penicillin/streptomycin (Gibco 15140). For resetting experiments, PER2::LUC rhythms were monitored using an Actimetrics luminometer housed within an incubator set to 37°C. At times spanning the circadian cycle, GABA (200 μM) or the equivalent volume of DMEM was added to the culture medium after 3 days in vitro. The direction and magnitude of the resetting response was quantified using Lumicycle analyses software (Actimetrics) by measuring the difference between the predicted and actual time of peak PER2::LUC expression on the cycle

following drug/vehicle application. Derivatized GABA concentration over time in culture was quantified using high-performance liquid chromatography (HPLC) coupled to fluorescence detection. Briefly, media samples were diluted 200-fold with HPLC grade water prior to pre-column derivatization with ortho-phthalaldehdye (OPA) in the presence of 2-mercaptoethanol using a Shimadzu LC10AD VP autosampler. Chromatographic separation was achieved using a Kinetex XB C-18 (50 × 4.6 mm, 2.6 µm; Phenomenex) and a mobile phase consisting of 100 mM Na2HPO4, 0.1 mM EDTA, and 10% acetonitrile (pH of 6.04). GABA was detected using a Shimadzu 10RF-AXL fluorescence detector with an excitation and emission wavelength of 320 and 400 nm, respectively.

## Immunohistochemistry

Brains were collected at eight timepoints spanning the circadian cycle (n = 4/timepoint/photoperiod), and protein expression was evaluated in free-floating slices with triple immunohistochemistry using methods described previously (*Evans et al., 2015*). Briefly, SCN slices (40 µm) were washed six times in PBS, blocked in normal donkey serum, and incubated for 48 hr at 4°C with primary antibodies for AVP (1:1K, Peninsula Laboratories, Cat# T-5048, RRID: AB_2313978), NKCC1 (1:500, Abcam, Cat#ab99558, RRID:AB_10675276) and KCC2 (1:500, EMD Millipore, Cat# 07–432, RRID: AB_310611). Following primary antibody incubation, SCN slices were washed six times in PBS, incubated for 2 hr at room temperature with secondary antibodies (1:500 for each: RRID:AB_10893040, Alexa Fluor 488; RRID:AB_10894526, Alexa Fluor 555; RRID:AB_10895029, Alexa Fluor 647), washed six times in PBS, and then mounted onto microscope slides using Promount anti-fade gold reagent. Fluorescence images were obtained with a Nikon A1R+ confocal microscope using identical settings for all samples. Images were analyzed in ImageJ. Using AVP immunoreactivity, regions of interest (ROIs) were selected for each SCN slice by placing a circular ROI on the AVP+ dorsomedial shell and a second ROI of identical size and shape in the AVP-negative ventral core. To calculate chloride co-transporter expression, each ROI was transferred onto KCC2 and NKCC1 images thresholded for background subtraction. For each sample, the KCC2/NKCC1 ratio was calculated and averaged across the two SCN lobes. Similar results were obtained with free-form ROIs encompassing each SCN compartment, but this approach was discarded because AVP expression in the SCN core was higher and more variable across samples. Lastly, mean daily expression of each chloride co-transporter was calculated by collapsing data from all samples collected at different timepoints.

## PER2::LUC imaging and analyses

SCN PER2::LUC rhythms were imaged as in *Evans et al. (2013)*. Briefly, PER2::LUC was collected using a Stanford Photonics CCD camera within a light-tight incubator set to 37°C. For drug treatments, pharmacological agents were added at the start of the experiment directly to the culture medium (always <0.05% total volume of medium) and remained for the duration of the recording. SCN slices were treated with the NKCC1 antagonist Bumetanide (80 µM, BU), the KCC2 antagonist VU0240551 (80 µM, VU), the KCC2 agonist CLP290 (100 µM), the carbonic anhydrase antagonist Acetazolamide (ACE, 100 µM), or vehicle (DMEM with DMSO <0.1%). To maintain drug efficacy for long-term recordings in this interface-culture preparation, drug concentration was increased slightly relative to previous work employing bath application of drugs for short-term electrophysiological recordings (i.e., 50 µM BU [*Choi et al., 2008*], 75 µM VU [*Haam et al., 2012*], 50 µM ACE [*Ostroumov et al., 2016*]). To assess effects of drug treatment on whole SCN rhythms, the time series for the entire SCN was extracted using ImageJ and analyzed with Lumicycle analyses software.

To analyze SCN function at the network and cellular level, Matlab-based computational analyses were used as described previously (*Evans et al., 2013*; *Evans et al., 2011*). Briefly, a time series was generated for each 12-pixel diameter ROI of the image using a uniform grid with 2-pixel spacing, for which the linear trend was eliminated and a Butterworth filter was applied to remove high- and low-frequency interference. To generate composite phase maps illustrating results for an entire group, samples were aligned to the same X-Y coordinates by minimizing the sum of squared difference of the 24 h-summed bioluminescence profiles. To evaluate cellular rhythms, an iterative process was employed to locate and extract data from cell-like ROIs after background and local noise subtraction. To measure interactions among SCN shell and core neurons, we calculated the phase difference between peak PER2::LUC in SCN core ROIs and the average peak time of SCN shell ROIs over

Cycles 2–4 in vitro, as in *Evans et al. (2013)*. The change in the phase relationship over time in vitro was used to quantify the magnitude and direction of coupling responses. Cellular period was calculated as the average peak-to-peak cycle length over the same interval in vitro. For both coupling and period responses, the relative peak time was determined subtracting the peak time of each cell on Cycle two from the average peak time of the complementary cell type.

## Circadian behavior assays

For drug treatments in vivo, mice received bumetanide (BU, 5 mg/kg) or vehicle (DMSO) in their drinking water for 8 weeks of photo-entrainment (L12 or L20) and 5 weeks of constant darkness (DD). Bumetanide levels were tested in plasma and hypothalamus with a bumetanide ELISA Kit (Neogen Corporation, Cat#103719–1) according to manufacturer's instructions. Drug concentration was assessed with a VersaMax Microplate Reader (Molecular Devices) and compared to a standard curve. To assess effects of bumetanide on circadian behavior, daily locomotor rhythms were monitored via wheel-running cages and the Clocklab acquisition system (Actimetrics, Evanston, IL). Using ClockLab software (Actimetrics, Evanston, IL), the time of activity onset, activity offset, activity duration, and the number of wheel revolutions was determined for each day of the experiment. The time of activity onset was identified each day as the first bin above a threshold of 15 counts, preceded by at least 6 hr of inactivity and followed within 30 min by at least two more bins likewise above threshold. Activity offset was determined by a similar but opposite rule. The difference between the time of activity offset and onset was used to calculate activity duration. Free-running period was measured from the slope of a regression line fit to activity onsets over the four weeks in DD. Lastly, effects of BU on jet-lag behavior was assessed in a subset of mice by abruptly advancing the L12 light:dark cycle by 6 hr. Rate of re-entrainment was quantified by calculating the number of days required for each mouse to shift activity onset by 6 hr, at which point the activity rhythm was re-aligned to the new LD cycle.

## Statistical analyses

Data are represented in figures as Mean ± SEM. Statistical analyses were performed with JMP software (SAS Institute, Cary, NC) or CircWave software (*Oster et al., 2006*). Phase, coupling, and period response curves were generated using a 4 hr running average, then collapsed into time bins for statistical analyses. Full Factorial ANOVA was used to assess effects of photoperiod and circadian time (or drug condition or cell type), as well as the interaction of the two main variables. Post hoc pairwise comparisons were performed with Least Square Means Contrasts to correct for family wise error. Changes in circadian behavior over time and its interaction with photoperiod and drug condition were analyzed with Full Factorial Repeated Measures ANOVA, followed by post hoc Least Square Means Contrasts. Statistical significance was set at $p \leq 0.05$ in all cases.

## Acknowledgements

We would like to thank Stanford Photonics and Adriano DellaPolla for technical assistance. Funding from the NIH (R01NS091234) and the Whitehall Foundation (2014-12-65) supported this work.

## Additional information

### Funding

| Funder | Grant reference number | Author |
| --- | --- | --- |
| National Institutes of Health | R01NS091234 | Jennifer Evans |
| Whitehall Foundation | 2014-12-65 | Jennifer Evans |

The funders had no role in study design, data collection and interpretation, or the decision to submit the work for publication.

### Author contributions

Kayla E Rohr, Harshida Pancholi, Nicholas J Raddatz, Formal analysis, Investigation, Methodology, Writing—review and editing; Shabi Haider, Christopher Karow, David Modert, Formal analysis,

Investigation, Writing—review and editing; Jennifer Evans, Conceptualization, Data curation, Formal analysis, Supervision, Funding acquisition, Investigation, Visualization, Methodology, Writing—original draft, Project administration, Writing—review and editing

#### Author ORCIDs
Jennifer Evans (iD) https://orcid.org/0000-0002-6220-0273

#### Ethics
Animal experimentation: This study was performed in strict accordance with the recommendations in the Guide for the Care and Use of Laboratory Animals of the National Institutes of Health. All of the animals were handled according to approved institutional animal care and use committee (IACUC) protocols (#AR-282) of Marquette University.

#### Decision letter and Author response
Decision letter https://doi.org/10.7554/eLife.49578.022
Author response https://doi.org/10.7554/eLife.49578.023

## Additional files

#### Supplementary files
• Supplementary file 1. Statistical analyses for GABA resetting, chloride co-transporter expression, and behavioral results.
DOI: https://doi.org/10.7554/eLife.49578.018

• Supplementary file 2. Statistical analyses for coupling and cellular period responses.
DOI: https://doi.org/10.7554/eLife.49578.019

• Transparent reporting form DOI: https://doi.org/10.7554/eLife.49578.020

#### Data availability
All data generated or analyzed during this study are included in the manuscript and supporting files. Data files have been provided for Figures 3, 4, and 6.

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
