## [Decision Letter]

Thank you for submitting your article "Seasonal plasticity in GABA A signaling is necessary for restoring phase synchrony in the master circadian clock network" for consideration by *eLife*. Your article has been reviewed by two peer reviewers, and the evaluation has been overseen by a Leslie Griffith as Reviewing Editor and Eve Marder as the Senior Editor. The reviewers have opted to remain anonymous.

The reviewers have discussed the reviews with one another and the Reviewing Editor has drafted this decision to help you prepare a revised submission.

Summary:

This paper addresses the interesting issue of how changes in light, as occur over seasons, affect the SCN. The authors find that GABA responsiveness changes in a reversible way with exposure to "long-day" light conditions. These changes revert back to the baseline in slices from animals adapted to summer conditions and this is correlated with changes in Cl^-^ gradients. The work has important implications for how coupling of SCN networks is regulated on a seasonal time scale.

Essential revisions:

Despite the enthusiasm of both reviewers, the reviewers had several concerns with the presentation of the data and statistical analyses which limit the interpretability of the data. These concerns are all related to analysis so should not require new experiments.

1) There are no statistical analyses presented to support the authors' statements that the rate of core/shell resynchronization is altered by any of the pharmacological treatments in Figure 4 and Figure 6. The authors should include an analysis of the core/shell phase difference across each cycle in culture, either by comparing the average across the whole core and shell or by examining only the cells in the polarized states (as in Figure 4D). Addition of this analysis would also greatly facilitate comparisons between this experiment and the in vivo experiment in Figure 5.

2) Using a ratio of KCC2/NKCC1 expression based on IHC fluorescence is questionable. While the authors are careful to not over interpret these data in the text of the results and acknowledge the limitations of IHC quantification in the Discussion, the way the data are represented in Figure 2 is still misleading (i.e. the label on the color bar in Figure 2B). The manuscript would be greatly improved by removing the KCC2/NKCC1 ratios entirely and instead focusing on the raw values for each transporter separately. An additional problem with the ratio analysis is that the data presented are not on a log scale, and the cosinor analysis run on log-transformed data because a floor effect is likely hiding a rhythm in the KCC2/NKCC1 ratio in the L12 shell. Furthermore, a 3 color gradient should be used because the differences on the red end of the scale are hard to tell apart.

3) Statistical reporting: It is unclear whether the vehicle treated slices were used in the analysis for the data in Figure 1D. If they were not, they should be included, and the responses to vehicle treatment for each photoperiod should be included in Figure 1D. The data in Figure 2D-E should not be analyzed with a t-test, but should be determined by looking for a main effect of photoperiod in the two-way ANOVA that was run for Figure 2—figure supplement 1C. A two-way mixed-design ANOVA should be used to analyze the activity duration for days in DD in the same way that it was for LD (Figure 5B). The Materials and methods should include details on the statistical tests used throughout and what (if any) assumptions of those tests were assessed. Details on any corrections for family-wise error should be stated. Clarification is needed on what is meant by "Full Factorial ANOVA omnibus test". Reporting the main effects of photoperiod, time and/or interactions would be more informative, along with f values and degrees of freedom. Sample sizes for individual timepoints/bins in Figure 1 and for neurons in "polarized" state (in Figures 4D and 6C) should be reported.

4) Writing: In general, the manuscript is well written; however, the writing is better understood by someone in the field and there is a concern that a reader outside the circadian field would not understand or appreciate the findings. The Introduction should be more concise and highlight the gap in the literature more quickly. As it stands now the sentence describing the prior work in the lab (Evans et al., 2013) in the next to last paragraph of the Introduction sounds just like the sentence in the last Introduction paragraph "these results reveal.…" and thus the contribution sounds incremental. Also the authors should use more specific terms and avoid jargon.

---

## [Author Response]

Essential revisions:Despite the enthusiasm of both reviewers, the reviewers had several concerns with the presentation of the data and statistical analyses which limit the interpretability of the data. These concerns are all related to analysis so should not require new experiments.1) There are no statistical analyses presented to support the authors' statements that the rate of core/shell resynchronization is altered by any of the pharmacological treatments in Figure 4 and Figure 6. The authors should include an analysis of the core/shell phase difference across each cycle in culture, either by comparing the average across the whole core and shell or by examining only the cells in the polarized states (as in Figure 4D). Addition of this analysis would also greatly facilitate comparisons between this experiment and the in vivo experiment in Figure 5.

We have included statistical analyses for all the response curves (i.e., embedded tables below x axis in Figure 4A, 6B, and Figure 4—figure supplement 1), and we have added supplementary files with full statistical reports. In revised Figure 4—figure supplement 1, we include sample sizes for all time bins divided by each cell type.

2) Using a ratio of KCC2/NKCC1 expression based on IHC fluorescence is questionable. While the authors are careful to not over interpret these data in the text of the results and acknowledge the limitations of IHC quantification in the Discussion, the way the data are represented in Figure 2 is still misleading (i.e. the label on the color bar in Figure 2B). The manuscript would be greatly improved by removing the KCC2/NKCC1 ratios entirely and instead focusing on the raw values for each transporter separately. An additional problem with the ratio analysis is that the data presented are not on a log scale, and the cosinor analysis run on log-transformed data because a floor effect is likely hiding a rhythm in the KCC2/NKCC1 ratio in the L12 shell. Furthermore, a 3 color gradient should be used because the differences on the red end of the scale are hard to tell apart.

We have moved the raw data for KCC2 and NKCC1 expression to the main Figure 2. We have revised the color scale for the ratiometric images as requested. To better visualize the lack of rhythm in the L12 SCN shell, we have plotted data for this compartment on a separate scale. From this revised figure, it is clear that a floor effect is not obscuring a KCC2/NKCC1 rhythm in this region. We have retained the ratio data since these proteins together are known to regulate chloride flux, and our predictions are based on the ratiometric model.

3) Statistical reporting: It is unclear whether the vehicle treated slices were used in the analysis for the data in Figure 1D. If they were not, they should be included, and the responses to vehicle treatment for each photoperiod should be included in Figure 1D. The data in Figure 2D-E should not be analyzed with a t-test, but should be determined by looking for a main effect of photoperiod in the two-way ANOVA that was run for Figure 2—figure supplement 1C. A two-way mixed-design ANOVA should be used to analyze the activity duration for days in DD in the same way that it was for LD (Figure 5B). The Materials and methods should include details on the statistical tests used throughout and what (if any) assumptions of those tests were assessed. Details on any corrections for family-wise error should be stated. Clarification is needed on what is meant by "Full Factorial ANOVA omnibus test". Reporting the main effects of photoperiod, time and/or interactions would be more informative, along with f values and degrees of freedom. Sample sizes for individual timepoints/bins in Figure 1 and for neurons in "polarized" state (in Figures 4D and 6C) should be reported.

In the revised manuscript, we have included analyses of vehicle responses, which do not differ by time of treatment or photoperiod (Figure 1—figure supplement 1, Full Factorial ANOVA, p > 0.4). As requested, data in Figure 2D and 2E are now analyzed with Full Factorial ANOVA, followed by paired Least Square Means contrasts – this has not changed our conclusions for this dataset. As requested, data in Figure 5B-C have been analyzed with Full Factorial Repeated Measures – this has not changed our conclusions for this dataset. We have removed reference to the ANOVA omnibus test, described our statistical tests in greater detail in the Materials and methods, clarified how we corrected for family wise error, added Supplementary files 1 and 2 with full statistical statements, and specified sample sizes for all response curves.

4) Writing: In general, the manuscript is well written; however, the writing is better understood by someone in the field and there is a concern that a reader outside the circadian field would not understand or appreciate the findings. The Introduction should be more concise and highlight the gap in the literature more quickly. As it stands now the sentence describing the prior work in the lab (Evans et al., 2013) in the next to last paragraph of the Introduction sounds just like the sentence in the last Introduction paragraph "these results reveal.…" and thus the contribution sounds incremental. Also the authors should use more specific terms and avoid jargon.

We have revised our Introduction as requested.